# Consumers' Attitude and Perception toward Traditional Foods of Northwest Greece during the COVID-19 Pandemic

**Dimitris Skalkos** [1,*] **, Ioanna S. Kosma** [1] **, Eleni Chasioti** [1] **, Adriana Skendi** [2] **, Maria Papageorgiou** [2] **and Raquel P. F. Guiné** [3]

1   Laboratory of Food Chemistry, Department of Chemistry, University of Ioannina, 45110 Ioannina, Greece; i.kosma@uoi.gr (I.S.K.); pch1241@uoi.gr (E.C.)
2   Department of Food Science and Technology, International Hellenic University, P.O. Box 141, GR-57400 Thessaloniki, Greece; andrianaskendi@hotmail.com (A.S.); mariapapage@food.teithe.gr (M.P.)
3   CERNAS Research Centre, Polytechnic Institute of Viseu, 3504-510 Viseu, Portugal; raquelguine@esav.ipv.pt
*   Correspondence: dskalkos@uoi.gr; Tel.: +30-265-100-8345

**Abstract:** Traditional foods (TFs) have a significant impact on the society and the economy of the areas where they are produced. The COVID-19 crisis, with the restrictions on daily living, is expected to cause a long-term influence on peoples' lives worldwide. This paper investigates the consumers' attitude and perception of TFs of northwest Greece in order to assess the possible impact of the pandemic toward the consumption of this kind of food. A questionnaire survey of self-response was carried out in fall 2020 on a sample of 510 participants through the Google platform. To analyze the data, basic descriptive statistical tools were used, combined with crosstabs and chi-square tests. The results revealed that the participants know the regional TFs well, continue to choose them due to a number of reasons, which include: the quality to price ratio, being local products with local raw materials, the nutritional properties, the social impact, as well as their positive impact to the regional economy and promotion. They buy them primarily from the supermarkets. They would recommend them to others, and they have increased their consumption during the pandemic, even though they consider their marketing inadequate, and they do not purchase them through the Internet yet. They believe that consumers in other regions of Greece would buy them if they had access to them. The foods of choice are traditional cheese and other dairy products, followed by wines, and aromatic herbs, which are the main regional TFs. These results indicate that the COVID-19 crisis has not interfered in consumers' attitudes and perceptions regarding TFs; therefore, they have the potential to expand and grow further in the future. In fact, they can play a vital role as major economic drivers in the post-COVID-19 era for the regional and local economies of Europe and elsewhere.

**Keywords:** traditional foods; COVID-19; knowledge; questionnaire survey; food supply chain; healthy food

## 1. Introduction

A year after World Health Organization (WHO) declared COVID-19 as a pandemic [1], the majority of countries around the world are still in the state of a health emergency, as the disease spread rapidly to the six continents. The lockdown imposed the closing of many educational institutions and workplaces as well as strict restrictions in what comes to traveling and socializing. On the other hand, homes were transformed into temporary offices and classrooms, and online work became a regular practice nowadays. However, people who work in the food supply chain do not have work from home option; hence, they need to keep their typical office and outside routines [2,3]. The food industry, as every industry and business sector in the world, expects to see how the COVID-19 outbreak will affect its business, manufacturing products that are essential for daily life. Keeping the food supply chain alive by proper supply management strategies is very important to meet consumers' demands [4]. Consumers normally do not think much about how the food

ends up on their table and how it has been produced. At the beginning of this global crisis, food became so essential that people were afraid of not being able to obtain it somehow, and therefore the demand for food has augmented so much that some store shelves were rapidly emptied. However, the food supply chains were able to respond to this challenge, with many supply chain actors, including farmers, producers, distributors, and retailers, working hard to renew shelves [3].

Key issues have been upraised in the food sector, along the whole food supply chain during the COVID-19 outbreak and beyond. Under these circumstances, people tend to follow healthier diets as a way of protecting and strengthening their immune systems [5]. Food safety has been given more attention in order to prevent the transmission of COVID-19 among primary producers, industrials, retailers, and consumers. Additionally, food security concerns were inflated due to the confinement and lockdown limitations. Food sustainability problems have emerged in the pandemic era [6]. During a pandemic situation, maintaining the regularity of the flow throughout the whole supply chain is imperative to prevent the food crisis and mitigate the negative impact on the economy at local as well as global scales [7]. In such situations, food supply chain facilities must pay more attention to the maintenance of employees' safety and health and adaptation of working conditions [8]. To avoid excessive increment in the prices of food commodities, protectionist policies must be contained. The supply chain should be able to face and outpace the challenging situation generated by the outbreak [8]. The stock markets have reacted with increased price volatility [9]. High price volatility was observed for products such as fertilizers and agrochemicals, as well as in the food distribution sector. However, the stocks of food retailers were subject to low price volatility. These patterns were also reflected in the companies' profits published in financial reports [9].

Initial insights into the behavioral changes in the pandemic situation reveal variations in the popularity of Internet searches related to food. The interest for restaurants decreased while it increased for cooking recipes and delivery services, reaching huge popularity, comparable or higher than Google searches related to health issues [10]. A reduction in shopping frequency was detected, but no changes were observed in the shopping location. People's concerns about diet decreased, but the interest in food products such as flour, bread, fruits, milk, and chicken (related to cooking, baking, and storage conditions) increased. In addition, concerns regarding food shopping, health care, and economics were identified [10]. This constant evolution of the situation might be reflected in further changes in the consumers' behavior and rationality, as there is a constant change in what governs their choices and practices [11]. Because of mandatory change of lifestyle, consumers have changed their habits and motivations behind them. The lockdown was indeed related to a notable modification of food choice motivations, with an increased focus on aspects such as weight control, health, natural content, ethical concerns, sensory appeal, and mood, while significantly decreasing the importance of convenience, familiarity, and price [12]. Especially for individuals with overweight and obesity, the lockdown has taken its toll on healthy dietary choices [13]. Because diet-related conditions are a major risk factor for being hospitalized for COVID-19, the pandemic may even further challenge public health. After this international crisis comes to an end, it is essential to strengthen the research activities to provide technical solutions aimed to improve short food supply chains and local production systems because, in possible even worse future menaces, they will represent a potential lifeline [14]. The crisis we are going through outlines that understanding the human usage of natural resources as a pillar of sustaining local communities is of utmost importance [15], and food production embeds one of these usages.

In this new global trend, TFs may be one of the alternative products that can satisfy new consumers' needs and requirements during and after this pandemic crisis. TFs constitute a separate category of the food market in Europe, representing a key role in the daily food intake [16]. They have played a historically important role in the traditions of different cultures and regions, contributing to their sense of identity and pride [17]. They reflect cultural inheritance and have left their imprints on contemporary dietary patterns [18].

They are key elements for the dietary patterns in different countries and consequently are important to accurately estimate population dietary intakes. The European Commission provided the following definition of the term "traditional" related to foods in 2006: "Traditional means proven usage in the community market for a time period showing transmission between generations; this time period should be the one generally ascribed as one human generation, at least 25 years" [19]. Apart from the commercial interests, the definition of the term "traditional" is an important issue addressed through Europe [20]. Trichopoulou et al. provided an elaborative definition, which includes statements about traditional ingredients, traditional composition, and traditional types of production and/or processing [18]. Among the European countries, a formal definition has been found for traditional food products from the Italian Ministry of Agriculture that defines TFP as "Agrifood products whose methods of processing, storage and ripening are consolidated with time according to uniform and constant local use" [21]. Although these definitions try to capture the different dimensions of the concept of traditional food products, there is still one perspective that is still missing, namely a definition of this concept seen from the consumer's point of view. TFs transform themselves in the interface between the consumers and producers, protecting cultural associations within a geographical area or with traditional practices [22]. TFs have geographically and traditional indicators in the European Union, which promote and protect the names of quality foodstuffs, their origin, and authenticity (e.g., PDO: protected designation of origin, PGI: protected geographical indication, TGI: traditional specialty guaranteed [23,24]). For a better understanding of TFs concept, statements about traditional ingredients, traditional composition, and traditional type of production and processes, the European Food Information Resource Network has developed separate guidelines [18,25]. Specific sensory attributes, gastronomic heritages, eating habits, and association with certain local areas are additional characteristics of TFs [26,27]. Overall, TFs can contribute to enhancing the destination attractiveness and, thus they can support local agro-economies [28]. In recent years, increasing market interest in TFs can be observed, both in Greece and in other countries such as Poland [29] and in the EU itself [30–32]. In fact, sales of protected products are worth more than €50 billion, representing around 6% of the sales value of the entire EU food and drink market. Wine accounts for more than half, around 56% of the value of the market for traditional and regional products, followed by agricultural and food products and spirits [33]. Southern European countries have a more traditional food character due to a greater market share of small companies and a better climate, which supports a more widespread availability of TFs [34]. France has the largest market share, followed by Italy. Other significant producers include Spain, Greece, and Austria. A recent study has shown that TFs have to claim their place within this digitized landscape, grown rapidly in the COVID-19 era and beyond, by finding the balance between old and new, between preferences for food taste passed through generations and new lifestyles at 5G speed [35].

Currently, in the EU, Council Regulation (EC)1151/2012 on quality schemes for agricultural products and foodstuffs aims to help producers of these products to communicate the product characteristics and farming attributes to buyers and consumers by establishing voluntary quality schemes. The definition of the term "traditional" in the above document means proven usage on the domestic market for a period that allows transmission between generations; this period is to be at least 30 years. As an EU member state, Greece incorporated the provisions of the Regulation in the national Legislation with Ministerial Decree (3321/145849) issued by the Hellenic Ministry of Agricultural Development and Food. Furthermore, a system of checks at all stages of production, processing, and distribution of geographical indications and traditional specialties guaranteed was established and is being implemented by the Hellenic Agricultural Organization-Demeter (AGROCERT). Greece has registered 79 protected designation of origin (PDO) products out of a total of 661 [36]. The majority of the Greek PDO (27) belong to the class "fruit, vegetables, and cereals fresh or processed". There are 34 Greek products designated with a protected geographical indication (PGI) out of 881 in the register. Surprisingly, there is no designa-

tion of traditional specialty guaranteed (TSG) for any product in Greece, despite the vast variety of traditional products and food recipes. It is worth mentioning the small number of TSGs in the register (only 64). As a result, the market of products that bear the term "traditional" is not adequately regulated despite their share in the market also connected with the tourism sector. Producers of traditional products conforming with the provisions of Council Regulation (EC)1151/2012 need obviously stronger incentives to apply for the quality scheme of TSG or are disappointed by the bureaucracy behind it.

The aim of this work was to assess the factors associated with the consumption and preference for TFs, within the COVID-19 pandemic period, in the Greek northwest region of Epirus. Even though there is adequate research related to consumers' behavior toward TFs, this is the first study in Europe that explores consumers' motives within the current health crisis, since it was conducted at the end of 2020, and there were no other related publications published by that time. The region of Epirus was chosen because it produces a significant number of local TFs, such as a typical southern European region, which shows a preference for its own foods [34]. It is a mountainous region, isolated from the rest of Greece until 10 years ago due to the lack of proper highways, with a long history of local traditional food production such as traditional green pies used for the feeding of its own residences. Livestock with sheep and goats has been developed as well as primary self-employment by the regional farmer for centuries producing milk, which has been used for cheese and other dairy TFs products. To accomplish the objective, the current study examines the following consumers' motivations, attitudes, and perceptions:

(I)    Consumers' motivation and behavior toward food purchase:

First, participants' choice of food products in the purchase process is examined in order to evaluate the consumers' consciousness in choosing food product in the purchase process

(II)    Consumers' perception of the TFs of Epirus:

Participants' perception of Epirus' TFs in terms of authenticity, quality, health, locality, and regional socioeconomic impact are examined following findings of previous reports

(III)   Consumers' perception about the marketing of Epirus TFs:

Participants' evaluation for the way Epirus' TFs are promoted and marketed currently is examined in view of the coming post-COVID-19 era.

(IV)   Consumers' access to TFs of Epirus:

Participants' preferences in purchasing TFs of Epirus from the existing retail locations is examined following reported recent data

(V)    Consumers' evaluation of TFs of Epirus:

Finally, the participants' overall evaluation of Epirus' TFs they buy and consume in terms of the kinds they chose, their satisfaction, and their intention to continue purchasing them within the pandemic period and beyond is examined.

## 2. Materials and Methods

### 2.1. Data Collection and Sample Characterization

This survey was based on a questionnaire, which was developed to investigate the motivations that influence food choices and consumers' eating habits concerning the TFs of Epirus. According to the objective of the study, the questionnaire was built up in six parts. Each question was created in such a way that it could provide the best possible information for each section. The way the parts were built up was based on similar previous studies [23,37]. The first part included questions concerned with the social-demographic characteristics of the respondents and specifically, sex, age, level of education, civil state job situation, and permanent residency. The second part consisted of five questions and was designed to assess the motivations that lead the participants to purchase food. Issues such as quality and price ratio, convenience in supplying and preparing the food of choice, sales, and motivations led by advertisements, were taken into consideration. The third part

included seven questions focused on the general perception of the participants regarding the TFs of Epirus. Issues such as general awareness regarding the types, production, materials, nutritional benefits, reasons to buy, and contribution to the general promotion of the region were addressed. In the fourth part, issues concerning the marketing of TFs were assessed; three questions were given in order to evaluate the participants' overall perception of the ability to promote and the marketing strategies followed by the local companies. The fifth part included two questions that approached the buying behavior of the participants in relation to the TFs. Finally, in the sixth part, from a set of 5 questions, participants were asked to respond and provide information on the types of products they consume, their views on quality and price, their intention to increase or not the consumption in the future, and especially whether they increased the consumption of TFs due to the pandemic. To guarantee the quality of the data obtained through the application of the questionnaire, this was pretested with 30 respondents. This phase was pivotal to ensure that the questions were clear and understandable so that respondents could answer them easily. The research was carried out using electronic questionnaires as it was easier to distribute and collect during the lockdown period. The distribution method chosen was by e-mail, as similarly performed in papers investigating consumer behaviors [38–40]. A snowball method was used in order to obtain a large number of participants [41]. Due to the pandemic and the lockdown restrictions, the survey was conducted via e-mail, so the sample included only participants that had Internet access. However, the sample's population is very well distributed as it included a wide range of ages, civil state, level of education, etc., who were familiar with the new technologies. A higher rate for female respondents has been observed by other papers as well [42–45], leading to the conclusion that women are more likely to respond to food-related surveys as they are primarily involved in the household organization. Furthermore, the sample used is representative, that is, people who are familiar with the concept of TF and therefore could provide reliable answers (in order to accurately describe their choice to buy these foods), which allowed us to specifically investigate an exact small part of the Greek population. The research questionnaire was created through the Google platform and the Google Forms function due to the ability to direct export of the results to an Excel sheet for further processing. The geographical context for the present study was primarily the region of Epirus, focusing mainly on Ioannina city, which is the city with the largest and most diverse population of the region, as it hosts the university in northwestern Greece. The town of Ioannina has approximately 120,000 residents, corresponding to half of the whole Epirus region population. The sample included students, among others, and through them, the questionnaire was made available to their families, friends, and acquaintances. Respondents received e-mails explaining the purpose of the research and the importance of their participation, while there was an attached link that led to the electronic form of the questionnaire. Responses were anonymous, and no personal information was collected or correlated with any of the responses to ensure the protection of participants.

The survey occurred during the period October–November 2020 and consisted of 510 participants (Table 1).

**Table 1.** Sociodemographic characterization of the sample.

| Variable | Groups | % |
|---|---|---|
| Sex | Male | 41.3 |
| | Female | 58.7 |
| Age | 18–25 | 25.3 |
| | 26–35 | 21.1 |
| | 36–45 | 18.4 |
| | 46–55 | 23.1 |
| | 55+ | 12.1 |
| Level of education | None/Primary school | 0.4 |
| | Secondary school | 0.2 |
| | High school | 10.5 |
| | University | 88.9 |
| Civil state | Single | 50.1 |
| | Married | 43 |
| | Divorced | 5.9 |
| | Widow | 1 |
| Job situation | Employed | 72.4 |
| | Unemployed | 5.6 |
| | Student | 22 |
| Permanent resident of the Epirus region | Yes | 43.4 |
| | No | 56.6 |

Of the 510 participants, 41.3% were males and 58.7% females, while 43.4% were permanent residents of the Epirus region and 56.6% were not, leading to a quite even distribution between participants. The majority of the participants were aged 18–25, 26–35, and 46–55 (25.3%, 21.1%, and 23.1%, respectively), while the other age groups were those least represented, i.e., groups 36–45 (18.4%) and 55+ (12.1%). Regarding the level of education, most of the participants had higher education (university, 88.9%), and only 0.6% had completed primary and secondary school, while the employment status category was dominated by employed (72.4%) participants.

### 2.2. Data Analysis

The exploratory analysis of the data was achieved through basic statistical tools. The survey was prepared in Greek and divided into six parts:

Part I. Sociodemographic data (see Table 1)
Part II. Motivation and consumer behavior for food purchase
Part III. Consumer's perception for the traditional food of Epirus
Part IV. Consumer's perception for the marketing of Epirus' traditional foods
Part V. Consumer access to the traditional food of Epirus
Part VI. Consumer's evaluation for traditional products of Epirus

The sociodemographic characteristics (six questions) were collected in the first part of the questionnaire (two dichotomous, one ordinal variable, and three nominal variables). According to the objectives of the present study, the second section of the questionnaire recorded the motivations and consumer behavior toward food purchases (five ordinal variables). The third (four ordinal variables, one dichotomous and one multiple choice question with each response considered as dichotomous variable) and the fourth (one categorical and two ordinal variables) parts were designed to explore the degree of consumers' awareness for the traditional products of Epirus and to reveal their opinion on the level of marketing of these products, respectively. The fifth (one dichotomous variable and one multiple choice question with each response considered as dichotomous variable) and sixth (four ordinal variables and one multiple choice question with each response considered as dichotomous variable) parts of the questionnaire aimed to collect the consumer's buying habits and the quality perception of traditional products of Epirus region.

The statistical processing of the data was performed using IBM SPSS Statistics for Windows (Version 25.0, IBM Corp. Armonk, NY, USA). The normality test was performed on the data before proceeding with the other statistical tests. Data obtained from the Likert scale were considered as ordinal values.

The nonparametric tests were used. A nonparametric chi-square test was performed to test the distribution of variables of each group and response based on the hypothesized equal proportions for each variable. The chi-square independence test was used to determine whether there is an association between variables. Post-hoc tests for the chi-square independence test were used. The pairwise comparisons (z-tests) for independent proportions followed by a Bonferroni correction were applied to the data. In order to measure the strength of association (when it is present between two variables), the Phi, Cramer's V, or Kendall's tau-b test. The Cramer's V coefficient was used to analyze the strength of the significant relations found between some of the variables at study. This coefficient ranged from 0 to 1 and can be interpreted as follows: $V \approx 0.1$ the association is considered weak, $V \approx 0.3$ the association is moderate, and $V \approx 0.5$ or over, the association is strong. Sociodemographic characteristics were considered as predictor variables that could affect the other responses of the questionnaire. In all the tests performed, the level of significance considered was 5% ($p < 0.05$).

## 3. Results

Table 2 presents the frequency of food purchase according to participants' motives and shows that the participants by far (78.6%) prefer to buy food with a good quality to price ratio. However, it was observed that only sometimes they choose food because of its convenience (49.5%), because it is on sale (58.9%), because it is easy to prepare (52%), or it is properly advertised (47.9%).

**Table 2.** Participants' motivations for food purchase.

| Questions | Answers (%) | | |
|---|---|---|---|
| | No | Yes | Sometimes |
| 1. I usually choose foods with a good quality/price ratio | 1.2 | 78.6 | 20.2 |
| 2. I choose the food I eat because it is convenient to get it | 15.9 | 34.6 | 49.5 |
| 3. I usually buy food that is easy to prepare | 26.9 | 21.1 | 52 |
| 4. I usually buy food that is on sale | 26.4 | 14.8 | 58.9 |
| 5. Running advertising campaigns increases my desire to consume certain foods | 41.8 | 10.2 | 47.9 |

The results of the chi-square test, presented in Table 3, showed that there were significant differences between motives for food purchase in terms of:

1. Good quality to price ratio food: only between sex ($x^2 = 14.220$, $p = 0.001$);
2. Purchasing convenience: only between age ($x^2 = 23.351$, $p = 0.003$), and civil state ($x^2 = 21.971$, $p = 0.001$);
3. Easily prepared food: only between age ($x^2 = 16.753$, $p = 0.033$), civil state ($x^2 = 20.806$, $p = 0.002$), job situation ($x^2 = 11.202$, $p = 0.024$), and permanent residency ($x^2 = 14.655$, $p = 0.001$);
4. Discount food: only between age ($x^2 = 45.970$, $p = 0.000$), civil state ($x^2 = 20.997$, $p = 0.002$), and job situation ($x^2 = 36.755$, $p = 0.000$);
5. Running advertising campaigns: only between permanent residency ($x^2 = 8.893$, $p = 0.012$).

**Table 3.** Associations between variables (A) value motivation for food purchase, (B) value perception for TFs of the Epirus region and the sociodemographic variables.

| | Sex | | | Age | | | Level of Education | | | Civil State | | | Job Situation | | | Permanent Resident | | |
|---|---|---|---|---|---|---|---|---|---|---|---|---|---|---|---|---|---|---|
| | $X^2$ * | $p$ ** | V *** | $X^2$ | $p$ | V | $X^2$ | $p$ | V | $X^2$ | $P$ | V | $X^2$ | $p$ | V | $X^2$ | $p$ | V |
| **A. Motivation for Food Purchase** | | | | | | | | | | | | | | | | | | |
| 1. I usually choose foods with a good quality/price ratio | 14.220 | 0.001 | 0.167 | | | | | | | | | | | | | | | |
| 2. I choose the food I eat because it is convenient to get it | | | | 23.351 | 0.003 | 0.151 | | | | 21.971 | 0.001 | 0.147 | | | | | | |
| 3. I usually buy food that is easy to prepare | | | | 16.753 | 0.033 | 0.129 | | | | 20.806 | 0.002 | 0.144 | 11.202 | 0.024 | 0.106 | 14.655 | 0.001 | 0.170 |
| 4. I usually buy food that is on sale | | | | 45.970 | 0.000 | 0.213 | | | | 20.997 | 0.002 | 0.144 | 36.775 | 0.000 | 0.191 | | | |
| 5. Running advertising campaigns increases my desire to consume certain foods | | | | | | | | | | | | | | | | 8.893 | 0.012 | 0.132 |
| **B. Perception for TFs of Epirus Region** | | | | | | | | | | | | | | | | | | |
| 1. Can you identify the traditional foods of Epirus? | | | | 29.214 | 0.000 | 0.240 | 11.672 | 0.009 | 0.152 | 18.875 | 0.000 | 0.193 | 15.397 | 0.000 | 0.175 | 40.228 | 0.000 | −0.282 (phi) |
| 2. Epirus produces many traditional foods (foods produced with local raw materials) | | | | 17.867 | 0.022 | 0.133 | | | | 14.796 | 0.022 | 0.121 | 13.557 | 0.009 | 0.116 | 26.453 | 0.000 | 0.228 |
| 3. The traditional products of Epirus offer nutritional benefits | 12.754 | 0.002 | 0.159 | 25.783 | 0.001 | 0.160 | 71.781 | 0.000 | 0.268 | | | | 13.028 | 0.011 | 0.114 | 22.601 | 0.000 | 0.212 |
| 4. Reasons to buy traditional products of Epirus (possible multiple choice) | | | | | | | | | | | | | | | | | | |
| 4.1. They are healthier than industrialized ones | | | | | | | | | | | | | | | | 5.193 | 0.023 | 0.101 (phi) |
| 4.5. They are produced by local companies and producers | | | | | | | | | | | | | 7.533 | 0.023 | 0.122 | 19.019 | 0.000 | 0.194 (phi) |
| 4.6. They are high quality foods | | | | 13.296 | 0.010 | 0.162 | | | | | | | | | | | | |
| 4.7. They have a social impact on the local communities | 6.142 | 0.013 | 0.110 (phi) | 24.570 | 0.000 | 0.220 | | | | 12.578 | 0.006 | 0.158 | 14.855 | 0.001 | 0.172 | 14.332 | 0.000 | 0.168 (phi) |
| 4.8. They have an economic impact on the local communities | | | | | | | | | | | | | 13.090 | 0.001 | 0.161 | 10.444 | 0.001 | 0.144 (phi) |
| 5. Do you think that consumers outside Epirus would choose Epirus' traditional products if they were available in their area? | | | | | | | 65.942 | 0.000 | 0.256 | 26.943 | 0.000 | 0.163 | 11.560 | 0.021 | 0.107 | 21.112 | 0.000 | 0.204 |
| 6. Do you consider that the purchase and consumption of traditional products of Epirus contributes to the promotion of the region of Epirus? | | | | | | | 37.452 | 0.000 | 0.193 | | | | | | | | | |
| 7. Would you recommend someone you know to consume traditional products of Epirus? | 6.731 | 0.035 | 0.115 | 20.048 | 0.010 | 0.141 | | | | 14.593 | 0.024 | 0.120 | 20.189 | 0.000 | 0.142 | 21.112 | 0.000 | 0.204 |

* chi-square test, ** level of significance of 5%: $p < 0.05$, *** Cramer's or Phi coefficient.

Table 4 presents the participants' perception for the TFs of Epirus and indicates that by far they can identify these products (84.8%), they know that the Epirus region produces a variety of TFs (86.2%), and they also are aware of their nutritional benefits (78.8%). As for the reasons why they purchase them, their first choice was the production with local raw materials (68.2%), followed by the production by local companies (63.1%), and the impact on the local economy (61.3%). Reasons such as being healthier than the industrialized foods (41.6%), free of chemicals (43.6%), or presenting superior organoleptic properties (46.9%) or quality (41.6%), and the social impact to local communities (36.1%) represented less than 50% of the responses. Furthermore, the participants expressed a strong opinion for the promotion of the region via the TFs (95.9%), the recommendation of Epirus' TFs to others (82.8%), and the potential for use by residents outside the region if they could easily access them (82.5%).

The results of the chi-square test, presented in Table 3, showed that there were significant differences between participants' perception for the TFs of Epirus in terms of:

1. Knowledge of TFs: for age ($x^2$ = 29.214, $p$ = 0.000), level of education ($x^2$ = 11.672, $p$ = 0.009), civil state ($x^2$ = 18.875, $p$ = 0.000), job situation ($x^2$ = 15.397, $p$ = 0.000), and permanent residency ($x^2$ = 40.228, $p$ = 0.000);

2. Local production of TFs: for age ($x^2$ = 17.867 $p$ = 0.022), Civil state ($x^2$ = 14.796, $p$ = 0.022), job situation ($x^2$ = 13.557, $p$ = 0.009), permanent residency ($x^2$ = 26.453, $p$ = 0.228);

3. Nutritional benefits of TFs: for sex ($x^2$ = 12.754, $p$ = 0.002), age ($x^2$ = 25.783, $p$ = 0.001), level of education ($x^2$ = 71.781, $p$ = 0.000), job description ($x^2$ = 13.028, $p$ = 0.011), and permanent residency ($x^2$ = 22.601, $p$ = 0.000);

4. Reasons to buy TFs:

   4.1. Healthier than industrial: only for permanent residency ($x^2$ = 5.193, $p$ = 0.023);

   4.2. Local production: only for job situation ($x^2$ = 7.533, $p$ = 0.023) and permanent residency ($x^2$ = 19.019, $p$ = 0.000);

   4.3. High quality: only for age ($x^2$ = 13.296, $p$ = 0.010);

   4.4. Impact on local community: for sex ($x^2$ = 6.142, $p$ = 0.013), age ($x^2$ = 24.750, $p$ = 0.000), civil state ($x^2$ = 12.578, $p$ = 0.006), job situation ($x^2$ = 14.855, $p$ = 0.001), and permanent residency ($x^2$ = 14.332, $p$ = 0.000);

   4.5. Economic impact locally: for job situation ($x^2$ = 13.090, $p$ = 0.001), and permanent residency ($x^2$ = 10.444, $p$ = 0.001).

5. Purchase by residents outside Epirus: for level of education ($x^2$ = 65.942, $p$ = 0.000), civil state ($x^2$ = 26.943, $p$ = 0.000), job situation ($x^2$ = 11.560, $p$ = 0.021), and permanent residency ($x^2$ = 21.112, $p$ = 0.000);

6. Promotion of the Epirus region: only for level of education ($x^2$ = 37.452, $p$ = 0.000);

7. Recommendation of TFs to others: for sex ($x^2$ = 6.731, $p$ = 0.035), age ($x^2$ = 20.048, $p$ = 0.010), civil state ($x^2$ = 14.593, $p$ = 0.024), job situation ($x^2$ = 20.189, $p$ = 0.000), and permanent residency ($x^2$ = 21.112, $p$ = 0.000).

**Table 4.** Participants' perception of the TFs of Epirus.

| Questions | Answers (%) | | |
|---|---|---|---|
| | **No** | **Yes** | **I Don't Know** |
| 1. Do you know the traditional foods of Epirus? | 15.2 | 84.8 | |
| 2. Epirus produces many traditional foods (foods produced with local raw materials) | 1.4 | 86.2 | 12.4 |
| 3. The traditional products of Epirus offer nutritional benefits | 1.4 | 78.8 | 19.8 |
| 4. Reasons to buy traditional products of Epirus (possible multiple choice) | | | |
| 4.1. They are healthier than industrialized ones | 58.4 | 41.6 | |
| 4.2. They are free of chemical additives | 56.4 | 43.6 | |
| 4.3. They have superior organoleptic properties (taste. aroma etc.) | 53.1 | 46.9 | |
| 4.4. They are made with local raw materials | 31.8 | 68.2 | |
| 4.5. They are produced by local companies and producers | 36.9 | 63.1 | |
| 4.6. They are high quality foods | 58.4 | 41.6 | |
| 4.7. They have a social impact on the local communities | 63.9 | 36.1 | |
| 4.8. They have an economic impact on the local communities | 38.7 | 61.3 | |
| 5. Do you think that consumers outside Epirus would prefer Epirus' products if they are reachable in their area? | 0.8 | 82.5 | 0.8 |
| 6. Do you consider that the purchase and consumption of traditional products of Epirus contributes to the promotion of the region? | 1.4 | 95.9 | 2.8 |
| | No | Yes | Maybe |
| 7. Would you recommend to someone you know to consume traditional products of Epirus? | 0.8 | 82.8 | 16.4 |

Table 5 presents the participants' perception of the marketing of Epirus' TFs. They do not consider the marketing of the products satisfactory (only 7.1% positive answers), and they are not convinced that the local companies are able to promote their products outside the region (54.4% returned "maybe" as an answer). Finally, they believe that there is a great opportunity for the TFs marketing using modern Internet advertisement (50.8%), followed by the TV spots (31.5%), leaflets (15.9%) and only by 1.9% via radio spots.

**Table 5.** Participants' perception for the marketing of Epirus' TFs.

| Questions | Answers (%) | | |
|---|---|---|---|
| | **No** | **Yes** | **Maybe** |
| 1. Do you think that the companies that produce traditional products of Epirus are capable to promote the products significantly outside Epirus? | 10.8 | 34.7 | 54.4 |
| 2. Do you consider the promotion of the traditional products of Epirus is satisfactory? | 49.8 | 7.1 | 43.1 |
| 3. If not which of the following ways would you like to be improved? | Television spots | Radio spots | Internet advertisements | Leaflets and other advertisements offered at the points of sale |
| | 31.5 | 1.9 | 50.8 | 15.9 |

The results of the chi-square test, presented in Table 6, showed that there were significant differences between participants' perception of the marketing of TFs in terms of:

1. Promotion by local companies of the TFs: only for sex ($x^2 = 11.555$, $p = 0.003$), and permanent residency ($x^2 = 6.939$, $p = 0.031$);
2. Satisfactory promotion of Epirus' TFs: only for level of education ($x^2 = 39.224$, $p = 0.000$), and permanent residency ($x^2 = 9.392$, $p = 0.009$).

**Table 6.** Associations between variables (A) marketing of TFs, (B) access to TFs, and (C) evaluation of TFs of Epirus and the sociodemographic variables.

| | Sex | | | Age | | | Level of Education | | | Civil State | | | Job Situation | | | Permanent Resident | | |
|---|---|---|---|---|---|---|---|---|---|---|---|---|---|---|---|---|---|---|
| | X² * | *p* ** | V *** | X² | *p* | V | X² | *p* | V | X² | *p* | V | X² | *p* | V | X² | *p* | V |
| **A. Marketing of Epirus' TFs** | | | | | | | | | | | | | | | | | | |
| 1. The companies that produce traditional products of Epirus are able to promote the products significantly outside Epirus? | 11.555 | 0.003 | 0.151 | | | | | | | | | | | | | 6.939 | 0.031 | 0.117 |
| 2. Is the way of promoting the traditional products of Epirus is satisfactory? | | | | | | | 39.224 | 0.000 | 0.198 | | | | | | | 9.392 | 0.009 | 0.136 |
| **B. Access to Epirus' TFs** | | | | | | | | | | | | | | | | | | |
| 1. How often do you buy or consume traditional food of Epirus? | | | | 58.998 | 0.000 | 0.171 | | | | 37.410 | 0.000 | 0.157 | | | | 159.615 | 0.000 | 0.563 |
| 2. Where do you buy traditional products of Epirus from? (possible multiple choice) | | | | | | | | | | | | | | | | | | |
| 2.1. Supermarket | 5.806 | 0.016 | −0.107 (phi) | | | | | | | | | | | | | | | |
| 2.2. Grocery store | | | | 11.299 | 0.023 | 0.149 | | | | | | | 6.860 | 0.032 | 0.117 | 23.765 | 0.000 | 0.217 (phi) |
| 2.3. Independent stores | | | | 29.011 | 0.000 | 0.239 | | | | | | | | | | 36.874 | 0.000 | 0.270 (phi) |
| 2.4. Restaurants | 8.266 | 0.004 | 0.127 (phi) | | | | | | | | | | | | | | | |
| 2.5. Internet direct online | 6.878 | 0.009 | 0.116 (phi) | | | | 11.174 | 0.011 | 0.149 | | | | | | | | | |
| **C. Evaluation of the Epirus' TFs** | | | | | | | | | | | | | | | | | | |
| 1. What types of products of Epirus do you consume? (possible multiple choice) | | | | | | | | | | | | | | | | | | |
| 1.1. Cheese | | | | 12.483 | 0.014 | 0.157 | | | | | | | | | | 8.657 | 0.003 | 0.131 (phi) |
| 1.2. Wine | | | | 39.410 | 0.000 | 0.279 | | | | 9.471 | 0.024 | 0.137 | | | | 53.653 | 0.000 | 0.327 (phi) |
| 1.3. Pasta | | | | 9.571 | 0.048 | 0.138 | 9.124 | 0.028 | 0.135 | | | | | | | 19.267 | 0.000 | 0.196 (phi) |
| 1.4. Honey | | | | 40.996 | 0.000 | 0.285 | | | | 13.556 | 0.004 | 0.164 | 9.987 | 0.007 | 0.141 | 115.000 | 0.000 | 0.478 (phi) |
| 1.5. Dairy products | | | | 21.237 | 0.000 | 0.205 | 13.216 | 0.004 | 0.163 | | | | | | | 26.609 | 0.000 | 0.230 (phi) |
| 1.6. Oil | 3.848 | 0.050 | 0.087 (phi) | 10.088 | 0.004 | 0.141 | | | | | | | | | | 42.841 | 0.000 | 0.292 (phi) |
| 1.7. Herbs | | | | 16.687 | 0.002 | 0.182 | | | | | | | | | | 46.730 | 0.000 | 0.305 (phi) |
| 1.8. Legumes | | | | 17.296 | 0.002 | 0.185 | 8.223 | 0.042 | 0.128 | 9.348 | 0.025 | 0.136 | 10.327 | 0.006 | 0.144 | 12.086 | 0.001 | 0.155 (phi) |
| 1.9. Other | | | | | | | 9.368 | 0.025 | 0.137 | | | | | | | | | |
| 2. How satisfied are you with the packaging of the traditional products of Epirus? | | | | 20.994 | 0.050 | 0.118 | | | | | | | 12.585 | 0.050 | 0.113 | 11.931 | 0.008 | 0.155 |
| 3. How satisfied are you with the quality of the traditional products of Epirus? | | | | | | | | | | | | | 12.986 | 0.043 | 0.114 | | | |
| 5. I have increased the consumption of local and traditional foods within the pandemic | | | | 30.255 | 0.000 | 0.173 | 30.673 | 0.000 | 0.175 | | | | 23.305 | 0.000 | 0.153 | 17.179 | 0.000 | 0.185 |

* chi-square test, ** level of significance of 5%: *p* < 0.05, *** Cramer's or Phi coefficient.

Table 7 presents the participants' access to TFs of Epirus. The purchase of TFs varies among them: while the majority buys TFs at least once per week (38.9%), some buy daily (16.2%), and some at least once a month (26.7%). Only 15.6% buy TFs on special occasions, and a very small percentage of 2.5% never consume TFs. As for the place of purchase, supermarkets are by far the shops of preference for TFs (89.2%), followed by grocery stores (38.6%), independent stores (25.5%), restaurants (6.7%), and only a small percentage of 4.3% through the Internet.

The results of the chi-square test, presented in Table 6, showed that there were significant differences between participants' perception for access to TFs in terms of:

1. Frequency of TFs purchase and use: only for age ($x^2$ = 58.998, $p$ = 0.000), civil state ($x^2$ = 37.410, $p$ = 0.000), and permanent residency ($x^2$ = 159.615, $p$ = 0.000);
2. Purchase of TFs from supermarkets: only for sex ($x^2$ = 5.806, $p$ = 0.016);
3. Purchase from grocery stores: for age ($x^2$ = 11.299, $p$ = 0.023), job situation ($x^2$ = 6.860, $p$ = 0.032), and permanent residency ($x^2$ = 23.765, $p$ = 0.000);
4. Purchase from independent stores: for age ($x^2$ = 29.011, $p$ = 0.000), and permanent residency ($x^2$ = 36.874, $p$ = 0.000);
5. Consumption of TFs in restaurants: only for sex ($x^2$ = 8.266, $p$ = 0.004);
6. Purchase through Internet: only for sex ($x^2$ = 6.878, $p$ = 0.009), and level of education ($x^2$ = 11.174, $p$ = 0.011).

**Table 7.** Participants' access to Epirus' TFs.

| Questions | Answers (%) | | | | |
|---|---|---|---|---|---|
| 1. How often do you buy or consume traditional food of Epirus? | Never | On special occasions | At least once a month | At least once a week | Daily |
| | 2.5 | 15.6 | 26.7 | 38.9 | 16.2 |
| 2. Where do you buy traditional products of Epirus from? (possible multiple choice) | No | Yes | | | |
| 2.1. Supermarket | 10.8 | 89.2 | | | |
| 2.2. Grocery store | 61.4 | 38.6 | | | |
| 2.3. Independent stores | 74.5 | 25.5 | | | |
| 2.4. Restaurants | 93.3 | 6.7 | | | |
| 2.5. Internet direct online | 95.7 | 4.3 | | | |

Table 8 shows the overall evaluation and opinion of the participants about the TFs of Epirus. In relation to the types of Epirus' TFs, they consume mostly cheese, which was their first preference (94.3%), followed by other dairy foods such as milk, yogurt (69.4%), wine (39.1%), honey (38.5%), aromatic herbs (27.3%), pasta (19.6%), legumes (18.4%), and others (12.6%). The participants are satisfied with the packaging of the TFs (73.1%) and the quality of the products (64.1%), and they think that their prices are reasonable (80.6%). Finally, the majority of the participants have increased their consumption of Epirus' TFs during the pandemic period by 59.2%, while the rest kept the same level of purchase and consumption (40.8%).

The results of the chi-square test, presented in Table 6, showed that there were significant differences between participants' overall evaluation of the TFs in terms of:

1. Kind of foods they prefer:
   1. Cheese: for age ($x^2$ = 12.483, $p$ = 0.014), and permanent residency ($x^2$ = 8.675, $p$ = 0.003);
   2. Wine: for age ($x^2$ = 39.410, $p$ = 0.000), civil state ($x^2$ = 9.471, $p$ = 0.024), permanent residency ($x^2$ = 53.653, $p$ = 0.000);
   3. Pasta: for age ($x^2$ = 9.571, $p$ = 0.048), level of education ($x^2$ = 9.124, $p$ = 0.028), and permanent residency ($x^2$ = 19.267, $p$ = 0.000);

4. Honey: for age ($x^2$ = 40.996, $p$ = 0.000), civil state ($x^2$ = 13.556, $p$ = 0.004), job situation ($x^2$ = 9.987, $p$ = 0.007) and permanent residency ($x^2$ = 115.000, $p$ = 0.000);
5. Dairy products: for age ($x^2$ = 21.237, $p$ = 0.000), level of education ($x^2$ = 13.216, $p$ = 0.004) and permanent residency ($x^2$ = 26.609, $p$ = 0.000);
6. Olive oil: for sex ($x^2$ = 3.848, $p$ = 0.050), age ($x^2$ = 10.088, $p$ = 0.004), and permanent residency ($x^2$ = 42.841, $p$ = 0.000);
7. Aromatic herbs: for age ($x^2$ = 16.687, $p$ = 0.002), and permanent residency ($x^2$ = 46.730, $p$ = 0.000);
8. Legumes: for age ($x^2$ = 17.296, $p$ = 0.002) for level of education ($x^2$ = 8.223, $p$ = 0.042), civil state ($x^2$ = 9.348, $p$ = 0.025), job situation ($x^2$ = 10.327, $p$ = 0.006), and permanent residency ($x^2$ = 12.086, $p$ = 0.001);
9. Other foods: for level of education ($x^2$ = 9.368, $p$ = 0.025).

2. Packaging of TFs: for age ($x^2$ = 20.994, $p$ = 0.050), job situation ($x^2$ = 12.585, $p$ = 0.050), permanent residency ($x^2$ = 11.931, $p$ = 0.008);
3. Quality of TFs: only for job situation ($x^2$ = 12.986, $p$ = 0.043);
4. Increase consumption of TFs during the pandemic period: for age ($x^2$ = 30.255, $p$ = 0.000), level of education ($x^2$ = 30.673, $p$ = 0.000), job situation ($x^2$ = 23.305, $p$ = 0.000), permanent residency ($x^2$ = 17.179, $p$ = 0.000).

**Table 8.** Participant's evaluation of Epirus' TFs.

| Questions | Answers (%) | | | |
|---|---|---|---|---|
| 1. What types of products of Epirus do you consume? (possible multiple choice) | No | Yes | | |
| 1.1. Cheese | 5.7 | 94.3 | | |
| 1.2. Wine | 60.9 | 39.1 | | |
| 1.3. Pasta | 80.4 | 19.6 | | |
| 1.4. Honey | 61.5 | 38.5 | | |
| 1.5. Dairy products | 30.6 | 69.4 | | |
| 1.6. Oil | 85.4 | 14.6 | | |
| 1.7. Herbs | 72.7 | 27.3 | | |
| 1.8. Legumes | 81.6 | 18.4 | | |
| 1.9. Other | 87.4 | 12.6 | | |
| | Not at all | Not much | Very | Very much |
| 2. How satisfied are you with the packaging of the traditional products of Epirus? | 0.6 | 10.2 | 73.1 | 14.5 |
| 3. How satisfied are you with the quality of the traditional products of Epirus? | 0.6 | 1.8 | 64.1 | 32.9 |
| | Low | Average | High | |
| 4. What do you think about the price of traditional products of Epirus? | 1.0 | 80.6 | 16.7 | |
| | No | Yes | | |
| 5. I have increased the consumption of local and traditional foods within the pandemic | 40.8 | 59.2 | | |

## 4. Discussion

This is the first research conducted during the COVID-19 pandemic period studying consumers' attitudes and perceptions toward TFs in Greece. The study focuses on the long-run attitudes and perceptions of the customers. The region of Epirus in Greece was chosen for the study since it is a region with increased production and consumption of TFs. As shown in Table 1, the participants in the study, who were asked for their opinion about Epirus' TFs, were mainly permanent residents of the region (56.6%). The participants in the study were classified into age groups according to: 18 years $\leq$ age $\leq$ 25 years accounting

for 25.3%, 26 years $\leq$ age $\leq$ 35 years accounting for 21.1%, 36 years $\leq$ age $\leq$ 45 years accounting for 18.4%, 46 years $\leq$ age $\leq$ 55 years accounting for 23.1%, and $\geq$ 55 years accounting for 12.1%. As for the level of education, the majority of participants, 88.9% had a university degree. Regarding the civil state of the participants, 50.1% were single, and 43% were married. Regarding their job situation so far, 72.4% of the participants were employed, and 22% were students at the time of the study. Overall, the sociodemographic characteristics of the presenters showed a suitable distribution between the different categories, similar to other recent reports [35].

The participants' food choices are mainly driven by good quality to price ratio (78.6%), followed by the convenience to obtain it (34.6%), and easiness in the preparation (14.8%). Previous studies have shown that in 2015 the price was the most important food choice motivation in Portugal and Greece [46], while in 2019, the health and environmental issues were the main drivers in the Mediterranean countries [37]. It is worth mention that because of the current economic and health crisis, factors such as quality, easy access, and preparation of food may overcome from now on the other motivations. The results of the chi-square test, presented in Table 3, indicated that there were significant differences between: (a) "sex" regarding the quality to price ratio, with weak association (V = 0.167); (b) "age" regarding convenience, preparation, and on sale foods, with weak association (V = 0.151/0.129/0.213, respectively); (c) "civil state" regarding convenience, preparation, and on sale foods with weak association (V = 0.147/0.144/0.144); (d) "employment" regarding easiness in preparation and foods with a reduced price, with weak association (V = 0.106/0.191); and (e) "permanent residency" regarding preparation, and running campaigns of foods, with weak association (V = 0.170/0.132).

Overall, as shown in Table 4, the participants are well aware of the TFs, their nutritional benefits, and their significant kinds. They buy them primarily because of their local origin, local production), impact on the local communities), organoleptic properties, and naturalness (being free of chemicals and healthier when compared with the industrial counterparts. They believe that consumers outside the region would buy them if they had access to them and that TFs contribute to the promotion of the region of Epirus. Finally, they would recommend Epirus' TFs to others. Factors such as knowledge, local origin [47], taste, local impact [48], health benefits, and appropriateness [49] as consumers' perceptions for TFs have been reported in the past too. The results of the chi-square test, presented in Table 3, highlight the differences between: (a) "sex" regarding nutritional benefits, social impact to local community, and recommendation of TFs to other with weak association (V = 0.159/0.110/0.115, respectively); (b) "age" regarding knowledge of the TFs, kinds, nutritional benefits, purchase because of high quality foods, social impact to local communities, and recommendation to others with weak association (V = 0.240/0.133/0.160/0.162/0.220/0.141); (c) "level of education" regarding knowledge of TFs, nutritional benefits, preference by consumers outside the region, contribution to the regions' promotion with weak association (V= 0.152/0.268/0.256/0.193); (d) "civil state" regarding knowledge of TFs, variations, purchase because of local impact, preference by consumers outside the region, recommendation to others with weak association (V = 0.193/0.121/0.158/0.163/0.120); (e) "employment" regarding knowledge of TFs, kinds, nutritional benefits, purchase because of local production, social local impact, economic local impact, preference by consumers outside the region, recommendation to others with weak association (V = 0.175/0.116/0.114/0.122/0.172/0.161/0.107/0.142); and (f) "permanent residency" regarding knowledge of TFs, variety, nutritional benefits with moderate association (V = 0.282/0.228/0.212), purchase because they are healthier than the industrialized, produced by local companies, have social local impact and economic local impact, preference by consumers outside the region, recommendation to others with weak association (V = 0.101/0.194/0.168/0.144/0.204/0.204).

The results show, as indicated in Table 5, that participants believe that these products are not well promoted (49.8% no, and 43.1% not sure) and that the companies that produce them may not have the capacity to do so (10.8% negative answers). They suggest mainly

the Internet advertisement for proper promotion, followed by TV ads, and leaflets. These findings agree with reported data, which shows that most of the TFs' are small and medium enterprises (SMEs) that lack marketing strategies, even if a considerable proportion of them report good marketing capabilities that lead to a market orientation [50]. Moreover, SMEs with better performance check that their objectives are reached but do not compare their strategy with that of competitors [50]. Indeed, it is required to support the effective use of direct marketing strategies to improve marketing efficiency for the manufacturers of traditional foods [51]. A recent study indicates that in the after-COVID global economy, proper marketing activities of TFs should focus primarily on the image aspect, a place brand's link with the nature of the products, and showing their importance to the environmental preservation and social welfare of a specific region [52]. The results presented in Table 6 show significant differences between: (a) "sex" regarding promotion of TFs abroad with weak association (V = 0.151); (b) "level of education" regarding promotion in the region with weak association (V = 0.198); and (c) "permanent residency" regarding promotion in the region with weak association (V = 0.136).

As shown in Table 6, we observed significant differences between: (a) "sex" regarding purchase from supermarkets, restaurants, and e-commerce with weak association (V = 0.107/0.127/0.116); (b) "age" regarding the frequency of TFs purchase, purchase from the grocery store, independent stores with weak association (V = 0.171/0.149/0.239); (c) "level of education" regarding purchase via the Internet with weak association (V = 0.149); (d) "civil state" regarding the frequency of purchase with weak association (V = 0.157); (e) "job situation" regarding purchase from grocery stores with weak association (V = 0.117); and (f) "permanent residency" regarding the frequency of purchase with strong association (V = 0.563), purchase from grocery stores, and independent stores with moderate associations (V = 0.217/0.270). Our findings agree with reports of the past and during the COVID-19 crisis indicating the overwhelming capacity of the supermarkets against traditional local retailers for food purchase well documented for many years [10,53,54]. Recent studies indicate the increased preference for online purchase of foods during the pandemic [35,55], which was not clearly recorded in our findings (online purchase being only 4.3%).

The participants, as shown in Table 8, value the TFs of Epirus in decreasing order of preference: cheese, followed by other dairy products, wine, honey, herbs, pasta, legumes, and olive oil. They are satisfied with the packaging, the quality of the foods, and their prices. However, half of them are not sure whether they will increase consumption of TFs in the future. The findings in this work regarding the choice for traditional foods are associated directly with local food production. Out of the 113 Greek products that have been certified as PDO and/or PGI [41], Epirus counts four types of wine, four kinds of cheese (feta, galotyri, kefalograviera, and metsovone), one olive oil, and one table olive [42]. Cheese, namely the feta cheese, is by far the representative TF produced in the region by local dairy factories, as a brand name food exported all over the world. These dairy industries produce other foods as well, such as milk and yogurt, among others, which are consumed locally, and in the rest of Greece. Local wines and honey produced by small companies are the foods most consumed within the region. Recently, due to their health benefits, foods such as herbs and legumes are produced by local family companies for regional use also. Our findings in terms of consumers' perception of packaging, quality, price, and increased consumption of TFs are in agreement with similar findings for general healthy food preferences within the COVID-19 period [10,55]. As shown in Table 6, significant differences were registered between groups for: (a) "sex" regarding only the consumption of olive oil with weak association (V = 0.008); (b) "age" regarding consumption of cheese, wine, pasta, honey, dairy products, olive oil, herbs, and legumes with weak to moderate associations (V = 0.157/0.279/0.138/0.285/0.205/0.141/0.182/0.185), satisfaction and increased consumption in the future with weak association (V = 0.118/0.173); (c) "level of education" regarding consumption of pasta, dairy products, legumes, others, and increased future consumption with weak association (V = 0.135/0.163/0.128/0.137/0.175);

(d) "civil state" regarding consumption of wine, honey and legumes with weak association (V = 0.137/0.164/0.136); (e) "job situation" regarding consumption of honey, legumes, and satisfaction, quality, and increased future consumption (V = 0.113/0.114/0.153); (f) "permanent residency" regarding consumption of cheese, pasta, and legumes with weak association (V = 0.131/0.196/0.155) wine, honey, dairy products, olive oil, and herbs with moderate association (V = 0.327/0.478/0.230/0.292/0.305), satisfaction and increased future consumption with weak association (V = 0.155/0.185).

## 5. Conclusions and Future Research

The present study provides an initial insight into the behavioral attitude and perception of consumers toward the TFs, in a pandemic situation, an event that has not occurred with this intensity for the last 100 years. The present contribution outlined the meaning of TFs in the Greek consumers' minds, and it identified the variables that predicted the preference for the purchase of TFs during the pandemic and beyond. To this purpose, an online survey was applied to a sample of 510 persons with age and sex quotas balanced between ages and sex, educated and employed at the time of the survey. Because of the change in lifestyle, consumers are expected to change their habits and the motivations that drive their food choices and consumption. With people spending more at home and dining out becoming less accessible, we did not notice a major shift in people's attitudes and behaviors concerning TFs Our results show that the consumers participating in this study have not altered their positive attitude and perception toward the TFs of the northwest region of Greece (Epirus) due to the pandemic. They are consumers who evaluate good quality to price ratio, on sale and easy to prepare foods, and evaluate the TFs of Epirus as follows:

Regarding their perception of the TFs of Epirus, our findings indicate that they know that Epirus produces a lot of them, with nutritional benefits, made by local raw materials, and local food companies with an economic impact on the regional economy. They are fans of these products, recommending them to residents outside the region who, they believe, would purchase them if they were available in their areas. They believe that the TFs are a major promotion tool for the specific northwest region of Greece. However, they believe that TFs are not adequately promoted within the region and abroad, and this remains a major drawback for their growth and development, despite the cheap and easy access of the Internet advertisement, which is not used properly yet.

Regarding their buying habits, they buy them mainly from supermarkets or sometimes from grocery stores once per week or a month. The purchase online is still very low despite the tendency for more and more online shopping due to the pandemic worldwide. They buy the most popular kinds, namely the local traditional cheese and the other dairy foods, followed by wines and aromatic herbs. They are satisfied with the packaging, the quality, and the price of what they buy and consume, and they have increased the consumption during the pandemic.

These findings are promising for the role of the TFs, as major local economic drivers, after the COVID-19 crisis. However, further studies are needed to investigate the long-lasting effects and adaptation of TFs consumption behavior to the "new normality". These studies will also identify further the required parameters in order for the TFs to be integrated into the daily consumption of the consumers, thus strengthening their healthy diet.

As our sample was recruited by convenience, more women, more people with university degrees, and with jobs within the pandemic participated in the survey, and this constitutes one limitation of the study, also considering the relatively limited number of responses obtained. The second limitation of the study is the use of the TFs only of the region of Epirus as compared with all TFs of Greece and beyond. However, this study is the first approach to understand the habits of TFs purchase and consumption in the current pandemic crisis, highlighting which aspects are more relevant for the consumption of these types of products from the consumers' point of view.

Future studies should be directed in two different directions; the first studying all kinds of Greek TFs, either by themselves or by comparison with TFs of other countries in Europe or beyond, and the second studying the consumers' attitude and perception for Epirus' TFs in depth, looking at major parameters for Epirus' TFs choice such as trust of the food, i.e., taste, authenticity, health, sustainability, transparency, and safety, or trust to the food supply chain from farm to fork.

**Author Contributions:** Conceptualization, supervision, methodology D.S.; writing—original draft preparation, D.S. and I.S.K.; investigation, E.C.; formal analysis, software, A.S.; writing—review and editing, D.S., M.P. and R.P.F.G. All authors have read and agreed to the published version of the manuscript.

**Funding:** This research received no external funding.

**Informed Consent Statement:** Not applicable.

**Conflicts of Interest:** The authors declare no conflict of interest.

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
