# Peer review of "Consumers’ Attitude and Perception toward Traditional Foods of Northwest Greece during the COVID-19 Pandemic"

_applsci, doi:10.3390/app11094080_

Round 1

Reviewer 1 Report

The manuscript entitled “Consumers’ attitude and perception towards traditional foods of Northwest Greece during the COVID-19 pandemic” is dedicated to  assess possible impact of the COVID-19 pandemic towards the consumption of this kind of food. Author carried out a survey with a structured questionnaire by Google Platform. A sample of consumers (n=510) was involved and the statistical processing of the data was performed using IBM SPSS Statistics for Windows.In my humble opinion, the aim of this paper is interesting and worthy to investigate. The paper is structured as follows:

Introduction (it is clear and describes the subject treated in a good manner);

materials and methods (it provides some information on construct measurement and data collection);

results and discussion (it describes main findings and related comments);

conclusion (a few information is provided).

On the basis of the contents of the paper, some suggestions should be useful to improve its quality i.e.

- More literature should be provided to improve the contents of the discussion section that it is necessary;

- The aim of the paper and related research questions should be better structured, at the end of the introduction;

- The materials and methods section should be supported with a consistent literature i.e. author should explain more in-depth the phase of consumers’ interviews and provide references to justify the selected method;

- the description of the questionnaire should be expanded also inserting the questionnaire as an annex;

- authors should check the lines 340 and 418;

- I suggest to create a separate Discussion paragraph to improve the analysis comments and well-describe the final consideration that, currently, are not structured.

- lastly, please, consider the integration of another section dedicated to the limitation of the work e.g. the geographical area involved, and the future research.

On the basis of aforementioned comments, the present form of this paper is not eligible for the publication on IF journal as Sustainability. Some major revisions are due.

Reviewer 2 Report

This manuscript presents the results of Consumers' attitude and perception towards traditional foods of Northwest Greece during the COVID-19 pandemic. This study is related to other works and the authors emphasize its pioneering nature. The topic of the paper is interesting and important. The strengths of the paper are as follows: 1/ research material from a quantitative study. I suspect that it was CAWI method - Computer Assisted Web Interviews, although this was not explicitly stated by the authors - please comment on this argument. 2/ The important role of traditional food in developing the idea of sustainable food consumption and an attempt to explain and understand the complexity of consumer behavior.
The analysis is based on descriptive statistics, descriptive statistical tools with crosstabs and chi-square tests. The questionnaire was constructed with 6 parts and questions structured in different categories. Correct responses were obtained using Google Forms and analyzed using IBM SPSS Statistics for 199 Windows. The results indicate that there are differences with respect to selected sociodemographic characteristics. The paper is good in terms of topic and relevance to "sustainable consumption". However, it has shortcomings that require major improvements and more detailed comments.

Reviewer 3 Report

The paper is well structured. A range of relevant information has been collected and reported in a review, which relates to stated objectives and also demonstrates evidence of critical evaluation. Information selected is mainly appropriate and represents an adequate understanding of the field of inquiry.

The author correctly used the existing literature and well explained the contribution to literature.

I strongly suggest that the author should expand the last paragraph in introduction about aims. It includes the important information but I think need to be expanded. Additionally, the introduction author could briefly explain the outline of paper.

Authors used very well the methods and empirical analysis but it could be mentioned the sampling method used .

You should also explain the meaning of * in your tables.

The analysis of results is well explained but I think instead of having numbers you could write paragraphs with the explanations.

Conclusion is very short. You have few sentences for the main results, one sentence for the meaning, and 2 sentences for the further research. I would strongly recommend to expand it. You can explain further what those results mean, compare it with previous studies(as you did in previous sections).

Reviewer 4 Report

After reading the content of the reviewed paper "Consumers’ attitude and perception towards traditional foods of Northwest Greece during the COVID-19 pandemic” I must say that the issues undertaken are in line with the subject of the journal. The problem presented in the article is important and interesting, the research question has not been formulated clearly enough.

The paper is descriptive in the nature, there are only simple statistical models used in the paper, which is disadvantage of this study. 

Significant literature in the field is included in the research, although this part could be improved. The summary is not covering the whole idea of the  paper.

Round 2

Reviewer 1 Report

Dear Authors,

thank you for taking into account my suggestions and I appreciated your reply. The new version of your manuscript includes changes  and, currently, I believe that it would then be acceptable for publication in the Applied Sciences Journal.

Best regards.

Reviewer 2 Report

Manuscript Applsci-1180913-peer-review-v1 touches on the behavioral attitude and perception of consumers on traditional food markets, which is an alternative trend in the approach of how food can be developed for its sustainability. Consumers purchasing this type of food represent a new quality in society, which translates into higher production possibilities for traditional food and provides a further basis for the development of this market segment, sustainability of consumption, realization of the from farm to fork strategy. The current form of the article has been completed, the text has become more structured, logical and understandable, which has significantly improved the quality of the text.

There are errors in the recording of literature, including in item 29. 

Based on the above comments, I suggest accepting the article for publication.
